# High-Quality Preparation of Energy-Containing Microspheres with Cross-Scale Particle Size

**DOI:** 10.3390/mi16040416

**Published:** 2025-03-31

**Authors:** Jiang Liu, Hairui Bian, Guoqiang Yu, Jiachao Zhang, Yaozheng Wang, Dang Ding, Ning Sang, Fangsheng Huang

**Affiliations:** 1Shaanxi Institute of Applied Physical Chemistry, Xi’an 710061, China; liujiang@njust.edu.cn (J.L.); m17791841150@163.com (G.Y.); 2School of Chemistry and Chemical Engineering, Nanjing University of Science and Technology, Nanjing 210094, China; 3Technological Innovation Center of Functional Microcapsule, Hefei Zhongke Create-Micro Technology Co., Ltd., Hefei 230026, China; hairui_bian@163.com (H.B.); dd17352922663@163.com (D.D.); sangning_iat@163.com (N.S.); 4School of Instrument Science and Opto-Electronics Engineering, Hefei University of Technology, Hefei 230026, China; 18356514497@163.com (J.Z.); hfut_wyz@163.com (Y.W.); 5Institute of Advanced Technology, University of Science and Technology of China, Hefei 230088, China

**Keywords:** energy-containing microspheres, microfluidics, particle size regulation, dispersibility

## Abstract

Microfluidic granulation technology enables high-quality production of energy-containing microspheres, significantly enhancing both performance and safety. Although microfluidic methods allow control over microsphere particle size, the adjustment range remains limited; low yield and process discontinuity also restrict broader application in the synthesis of energy-containing materials. This paper presents a microfluidic granulation system for energy-containing materials utilizing pulsed pneumatic printing, co-flow, and flow-focusing techniques to achieve wide particle size adjustment, consistent particle formation, high granulation speed, and production efficiency. This system allows microsphere sizes between 110 and 2500 μm, with a coefficient of variation (CV) as low as 1.9%, a frequency exceeding 13,000 Hz, and a suspension consumption rate reaching 100 mL/h. Calcium alginate/potassium perchlorate microspheres, prepared with sodium alginate hydrogel as a binder, exhibit uniform structure, narrow size distribution, and efficient energy material loading. We anticipate further advancements in applying microfluidic technology to energy-containing microsphere production based on this system.

## 1. Introduction

Spherical energy-containing materials with high morphological consistency offer low sensitivity, optimal dispersion, and maximal packing density. Conventional spherization techniques—such as emulsion evaporation [1,2], cooling crystallization [3,4], and spray drying [5,6]—are simple in structure and process but often result in low sphericity, particle aggregation, and uneven particle sizes [7,8,9]. Microfluidic technology, by utilizing immiscible multiphase fluids in microchannels, allows for highly reproducible and flexibly regulated droplet formation with uniform particle distribution. With appropriate curing methods like ionic cross-linking, photopolymerization, and solvent evaporation, microfluidic systems can generate spherical products with regular shapes and consistent particle sizes [10,11,12]. Given these advantages, microfluidic technology is increasingly applied to the synthesis of energy-containing compounds, recrystallization and modification of monolithic materials, and preparation of composite energy-containing materials [13,14,15,16].

Current research on the microfluidic synthesis and production of energy-containing microspheres remains nascent, with key challenges still in high-throughput, integrated, and continuous production [17,18,19]. Although microfluidic technology offers distinctive advantages in particle size regulation, the adjustable range for microsphere size in current energy-containing applications remains limited [20,21,22]. Microsphere particle size directly influences performance: smaller particles, due to their large specific surface area, exhibit rapid energy release rates but may suffer from reduced thermal and mechanical stability, leading to heightened sensitivity and ease of external stimulation—suitable for high-efficiency energy release in propellants and high-performance explosives [23,24,25]. Conversely, larger microspheres typically offer improved thermal and mechanical stability, albeit with slower, more uniform energy release, making them more apt for industrial blasting and applications requiring consistent energy output [26,27]. Thus, in both production and applied research, optimization of microsphere size is crucial for achieving optimal performance and safety across varied applications.

This study leverages flow-focusing, co-flow, and pulse pneumatic printing technologies to develop an integrated microfluidic granulation system. The resulting microspheres exhibit high sphericity, narrow particle size distribution, and adjustable sizes ranging from 100 μm to 2500 μm. Through this microfluidic system, precise particle size control is achieved, enabling alignment with diverse application requirements and enhancing both preparation efficiency and performance of energy-containing materials. The broad cross-scale regulation further facilitates customizable microsphere production and enables detailed micro reaction analysis of energy-containing materials.

## 2. Experimentation and Characterization

### 2.1. Materials and Instruments

Materials: polyvinyl alcohol 1788, analytically pure AR, Hefei BASF Biotechnology Co., Ltd., Hefei, China; sodium alginate, analytically pure AR, Sinopharm Chemical Reagent Co., Ltd., Shanghai, China; calcium chloride, analytically pure AR, Sinopharm Chemical Reagent Co., Ltd., Shanghai, China; light-curing resin; industrial-grade potassium perchlorate, Jiangxi Yongning Technology Co., Ltd., Yichun, China; deionized water, homemade in the laboratory.

Instruments: microfluidic device, Hefei Zhongke Create-Micro Technology Co., Ltd., Hefei, China; precision syringe pump, LSP01-1A, Baoding Lange Constant Flow Pump Co., Ltd., Baoding, China; peristaltic pump, Hefei Zhongke Create-Micro Technology Co., Ltd., Hefei, China; programmable pulse gas printer, Hefei Zhongke Create-Micro Technology Co., Ltd., Hefei, China; magnetic stirring water bath; somatoscope.

### 2.2. Microfluidic Device (MFD)

As illustrated in Figure 1a–c and Table 1 (solution system), conversion among the three techniques is achieved by adjusting the nozzle morphology and the driving phase downstream of the system’s coaxial laminar flow. In the flow-focusing technique, the inner phase fluid is sheared by the driving phase and focused through small orifices to form a cone-jet, which breaks into droplets due to instability, producing microspheres with a minimum particle size of 110 μm. In the co-flow technique, the focused orifices are removed, and the jet disintegrates under the driving phase shear force, resulting in larger particles. In the pulse pneumatic printing technique, a gas valve generates a continuous pulsed gas flow as the driving phase, shearing the liquid at the capillary’s free end into uniform droplets.

### 2.3. Microsphere Preparation

Solution Preparation:

For the flow-focusing and co-flow methods, the dispersed phase consisted of a light-curing resin combined with 5 wt% potassium perchlorate, ultrasonically stirred for 30 min to ensure uniform particle dispersion within the suspension. For the continuous phase in both flow-focusing and co-flow techniques, polyvinyl alcohol (PVA) was added to deionized water and stirred magnetically at 80 °C for 15 min to yield a 2% aqueous PVA solution as the continuous phase.

In the pulse pneumatic printing technology, the dispersed phase was prepared by mixing sodium alginate and potassium perchlorate in deionized water, followed by heating and stirring in a magnetic stirring water bath at 90 °C for 30 min to obtain a suspension containing 2 wt% sodium alginate and 18 wt% potassium perchlorate. The driving phase consisted of a constant frequency pulsed gas generated by a programmable pulsed gas printer. The receiving liquid for the pulse pneumatic printing technology was an aqueous solution of calcium chloride, prepared by dissolving calcium chloride in deionized water to achieve a mass fraction of 2%.

During the experiment, Teflon tubing with an inner diameter of 2 mm was employed to connect the syringe to the inlet and outlet of the microfluidic chip. A precision syringe pump was utilized to propel the inner phase and driving phase solutions from the Teflon tube into the microfluidic chip’s inlet. Subsequently, microdroplets were generated at the chip’s outlet, and the resulting microspheres were cured through UV photopolymerization and ionic cross-linking downstream. The flow rates of both phases were meticulously controlled by adjusting their respective syringe pumps.

### 2.4. Characterize

The macroscopic morphology of the microspheres was characterized using an in situ optical microscope. Geometric parameters were extracted and the diameter and size distribution of the microspheres were quantified through image analysis of the optical microscope images utilizing ImageJ (version 1.54g; NIH, Bethesda, MD, USA).

## 3. Results and Discussion

### 3.1. Parameter Tuning and Morphology Analysis

During the preparation of microspheres utilizing flow-focusing and co-flow techniques, the flow rate ratio between the continuous phase and the dispersed phase serves as a critical factor influencing microcapsule size. An increase in the flow rate of the dispersed phase results in the generation of larger droplets, subsequently leading to the formation of larger microcapsules. Conversely, when the flow rate of the continuous phase is elevated, the droplets of the dispersed phase are more susceptible to shear, resulting in smaller droplets and, consequently, the production of smaller microcapsules.

As illustrated in the flow-focusing experiments presented in Figure 2a and Figure 3a, when the flow rate of the inner-phase photosensitive resin suspension was maintained at 30 mL/h, and the flow rate of the driving-phase PVA solution (Q*f*) was varied from 500 mL/h to 900 mL/h, the diameter of the produced microspheres systematically decreased from 315 μm to 110 μm. This observation indicates a negative correlation between the microsphere diameter and the driving-phase flow rate, aligning with the anticipated experimental outcomes.

In the co-flow experiments, as illustrated in Figure 2b and Figure 3b, the flow rate of the inner-phase photosensitive resin suspension was maintained at 5 mL/h. Meanwhile, the flow rate of the driving-phase PVA solution (Q*f*) was systematically increased from 300 to 1400 mL/h. Consequently, the diameter of the microspheres produced during this preparation process was progressively reduced from 905 μm to 270 μm.

In the microsphere preparation experiment utilizing the pulsed pneumatic printing technique, the frequency of microsphere generation was entirely regulated by the pulsed gas. In this study, the gas-driven frequency was set at 2 Hz, while variations in the inner-phase solution supply rate were employed to modulate the diameter of the generated microspheres. As depicted in Figure 2c and Figure 3c, the flow rate of sodium alginate was progressively increased from 7.5 mL/h to 100 mL/h, resulting in a corresponding increase in microsphere diameter from 860 μm to 2550 μm. The experimental findings indicate a positive correlation between microsphere diameter and the flow rate of the inner-phase solution at a constant pulsed gas frequency.

The microfluidic system developed in this study enables the precise fabrication of energy-containing microspheres ranging from micrometer to millimeter sizes, effectively addressing the requirements of diverse application scenarios regarding particle size. Smaller energy-containing microspheres, characterized by their large specific surface area, can significantly enhance reaction rates and combustion speeds, making them particularly suitable for propellants and high-performance explosives that demand high energy release efficiency. However, this increased reaction rate may also introduce challenges related to uneven energy release and compromised safety. Conversely, larger particle-sized energy-containing microspheres are more appropriate for industrial blasting and similar applications where stable energy output is essential, owing to their consistent energy release and enhanced safety profiles.

The microfluidic system developed in this study enables precise regulation of microsphere particle size to meet various application requirements, thereby significantly enhancing the preparation efficiency and performance of energy-containing materials.

### 3.2. Particle Size Distribution Coefficient

Particle size and size distribution are critical characteristics of microspheres, with narrower distributions and enhanced dispersion correlating to greater added value. Superior dispersion confers significant advantages to energy-containing microspheres across various applications, ensuring uniform energy release, minimizing localized overheating and safety risks, and increasing packing density and energy content. Furthermore, optimal dispersion enhances the flowability of microspheres, facilitating automated production and precise processing. In terms of mechanical stability, uniformly distributed microspheres are more resilient to stress, thereby reducing breakage and enhancing overall performance. In chemical reactions, effective dispersion increases the number of reaction interfaces, thereby improving reaction efficiency. Additionally, the average particle size and particle size distribution serve as essential indicators for assessing product repeatability. The formula for the number average particle size distribution coefficient (*CV*) is as follows:(1)$$CV=\frac{\sigma }{d}\times 100\%$$

In this context, *σ* represents the standard deviation of the particle size, while *d* denotes the mean particle size. The coefficient of variation (*CV*) is utilized to characterize the degree of dispersion within the particle size distribution. For instance, in the case of energy-containing microspheres produced through co-flow as depicted in Figure 2b, the flow rate of the inner phase was maintained at 5 mL/h, whereas the flow rate of the driving phase was set to 300 mL/h. The particle size distribution is illustrated in Figure 4, which includes a sample size of 103, a number average particle size of 905.24 μm, a standard deviation of 17.52 μm, and a *CV* value of 1.9%.

### 3.3. Microsphere Production Frequency

Pulse pneumatic printing technology actively regulates droplet formation during microsphere fabrication by precisely controlling the gas pulse frequency. In this process, the droplet generation frequency (*f*) is synchronized with the frequency of the gas pulses, allowing for precise modulation of the droplet formation rate through adjustments to the gas pulse frequency. This methodology offers a significant advantage by providing a highly consistent and controlled microsphere production process, thereby ensuring that the final product exhibits uniform size and morphology.

In contrast, the production frequency of microspheres fabricated using flow-focusing and co-flow techniques is influenced by a broader array of factors. Specifically, physical parameters including the dimensions of the capillary channel, the flow rates of the two-phase fluid, and the fluid’s viscosity significantly affect the microsphere production frequency. The interplay of these factors dictates the efficiency of droplet formation and, consequently, the overall microsphere production rate.

To provide a more precise description of the microsphere production process, a formula for the microsphere production frequency can be derived based on the inner phase flow rate and the anticipated particle size of the generated capsules.(2)$$f=\frac{Q_{i}}{V}=\frac{Q_{i}}{4/3\pi R^{3}}=\frac{Q_{i}}{1/6\pi d^{3}}=\frac{{6Q}_{i}}{\pi d^{3}}$$

As illustrated in Figure 5a,b, microspheres were synthesized using flow-focusing and co-flow techniques, with the generation frequency directly correlating to the flow rate of the driving phase (error bars represent standard deviation from three independent experiments). Notably, the microspheres were produced at an accelerated frequency due to the rapid movement of the polyvinyl alcohol (PVA) solution, which propels the fluid exiting the capillary tube and causes it to converge at a constricted aperture, forming a stable cone structure. At the apex of this cone, a fine microjet is generated, which subsequently disintegrates into monodisperse droplets owing to hydrodynamic instability. This focusing process, driven by fluid dynamics, facilitates the rapid production of droplets, as evidenced by the observed generation frequency of 12,289 Hz at an inner phase flow rate (*Q_i_*) of 30 mL/h and a driving-phase flow rate (*Q_f_*) of 900 mL/h. By meticulously regulating the frequency and size of microsphere production, it becomes feasible to ensure that the resulting energy-containing microspheres exhibit the requisite physical and chemical properties, thereby enhancing product performance and application efficacy.

As presented in Table 2, each of the three distinct technological pathways within the microfluidic granulation system developed in this study is associated with a specific range of particle size production. Flow-focusing is particularly suited for the synthesis of small-particle-size microspheres, exhibiting a rapid granulation frequency; however, this method demonstrates the least favorable dispersion. In contrast, co-flow is effective for producing sub-millimeter microspheres, characterized by excellent dispersion, albeit at a reduced production efficiency. Meanwhile, pulse pneumatic printing technology is optimized for generating millimeter-sized microspheres, which offers both good dispersion and high production efficiency, although it operates at a lower granulation frequency. Collectively, these three techniques enable the preparation of microcapsules ranging from the micrometer to the millimeter scale, thereby comprehensively addressing the production demands for more refined energy-containing microspheres.

## 4. Systematic Technical Integration Analysis

### 4.1. Core Nozzle Configurations

The three technologies—flow-focusing, coaxial flow-focusing, and pneumatic printing—share identical liquid supply components but differ primarily in their fluid outlet configurations. In flow-focusing, the inner needle aligns directly with a narrow focusing orifice in the driving chamber. In contrast, coaxial flow-focusing and pneumatic printing employ concentric alignment of the inner needle with a wider driving chamber outlet.

### 4.2. Focusing-Phase Mechanisms

The driving phase can be categorized as either a homogeneous liquid phase (liquid-driven) or a heterogeneous gas phase (gas-driven). Liquid-driven systems rely on stable liquid–liquid interfacial shear forces, which are further divided into flow-focusing and coaxial flow-focusing modes. Pneumatic printing, however, utilizes gas-driven shear, where gas–liquid interfacial interactions exhibit lower stability and weaker, more fluctuating shear forces compared to liquid–liquid systems. Notably, liquid-driven systems may introduce trace contamination due to material interactions between phases, whereas gas-driven systems minimize contamination by avoiding direct contact between the inner fluid and reactive media.

### 4.3. Droplet Formation Mechanisms

All three technologies require external driving forces to fragment the inner fluid into droplets:

Flow-focusing: droplets form via hydrodynamic instability of a slender jet in the focusing zone.

Coaxial flow-focusing: droplets are generated through steady shear-driven fragmentation at the inner needle outlet.

Pneumatic printing: droplets form at the capillary tip via gas-driven shear and gravitational forces.

### 4.4. Comparative Performance

The differences in nozzle geometry, driving phase, and fragmentation dynamics lead to distinct droplet characteristics under equivalent flow parameters:

Flow-focusing systems: Higher interfacial shear forces (due to velocity gradients between phases) result in higher droplet breakup frequencies and smaller sizes, albeit with reduced monodispersity. This technology is ideal for high-throughput production of submicron droplets.

Coaxial flow-focusing systems: Lower velocity gradients yield smaller interfacial shear forces, producing larger droplets with superior monodispersity, broader parameter tolerance, and isotropic properties. This method is optimal for applications requiring uniform droplet morphology.

Pneumatic printing systems: Gas–liquid interfacial instability and weaker shear forces, combined with gravitational effects, generate larger droplets with minimal contamination. This approach is suited for niche applications demanding large droplets or ultrapure formulations.

As shown in Figure 6 below, three droplet microfluidics techniques were integrated to build a device to realize cross-scale microdroplet preparation in energy-containing media.

The two-phase syringe pump supply module is used to provide the liquid-driven flow-focusing and the dispersed and driving phases in the co-flow system, respectively. The lower-left “peristaltic pump and buffer bottle” liquid supply module ensures excellent dispersion of raw materials in the process of high concentration energy-containing media droplet molding, and together with the one-phase pulsed gas flow generator module (lower-right), it can realize the quantitative supply of dispersed- and driving-phase media in pneumatic printing technology; together with the one-phase syringe pump, it can realize the quantitative supply of dispersed- and driving-phase media in the co-flow system. The dispersed phase and the driving phase can be dosed in a co-flow system by using a one-phase syringe pump.

In summary, the droplet mode is closely related to the role of the driving-phase flow. In general, the homogeneous liquid phase interfacial tension is smaller than the heterogeneous gas phase, so the gas-driven phase can prepare the largest template droplets; the homogeneous liquid-phase co-flow interfacial tension is smaller than the flow-focusing, so the liquid–liquid interfacial tension of the flow-focusing system can prepare the smallest droplet size, followed by the co-flow.

## 5. Summarize

Energy-containing material microspheres can be produced with high quality, controllable particle size, and excellent dispersibility through the microfluidic granulation process. However, the limited range of particle size regulation has restricted the broader application of microfluidic synthesis technologies for energy-containing materials as a general-purpose platform. This study successfully established a microfluidic granulation system characterized by an extensive range of particle size regulation and highly consistent particle formation properties by integrating pulse pneumatic printing technology, co-flow technology, and flow-focusing technology. Additionally, through a specialized microchannel structural design, the system achieved a consumption rate of energy-containing material/binder suspension of up to 100 mL/h, a granulation frequency of up to 12,289 Hz, a microsphere mass fraction of energy-containing material reaching 90%, and a diameter distribution coefficient of the microspheres of 1.9%. These advancements indicate promising prospects for the application of microfluidic technology in the synthesis and preparation of energy-containing materials, effectively addressing challenges related to high-throughput production and scalability.

## Figures and Tables

**Figure 1 micromachines-16-00416-f001:**
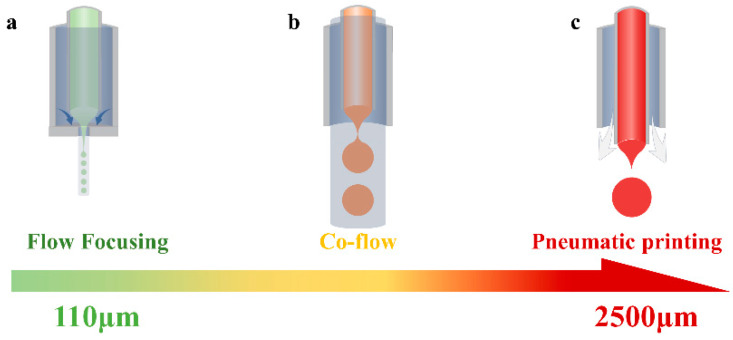
Three technology paths and corresponding microsphere size distributions. (**a**) Flow-focusing technique; (**b**) co-flow technique; (**c**) pneumatic printing technique.

**Figure 2 micromachines-16-00416-f002:**
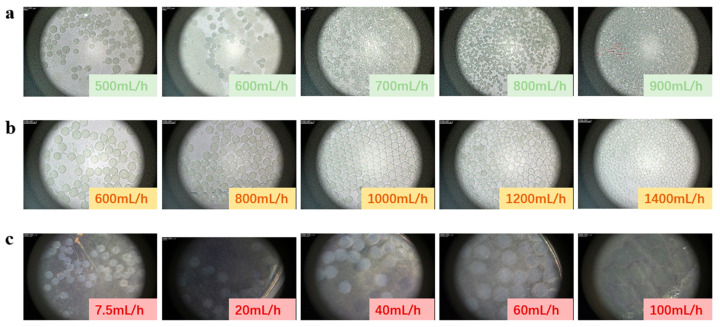
Energy-containing microspheres prepared by three techniques. (**a**) Flow-focusing technique; (**b**) co-flow technique; (**c**) pneumatic printing technique.

**Figure 3 micromachines-16-00416-f003:**
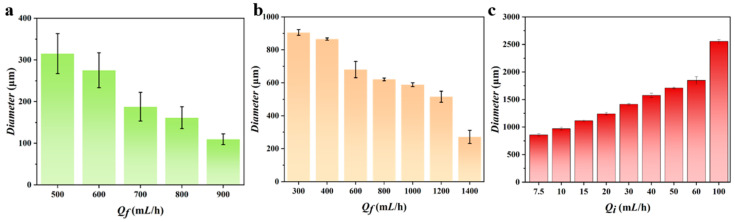
Particle size–flow relationship plots of microspheres prepared by three techniques. (**a**) Flow-focusing technique; (**b**) co-flow technique; (**c**) pneumatic printing technique.

**Figure 4 micromachines-16-00416-f004:**
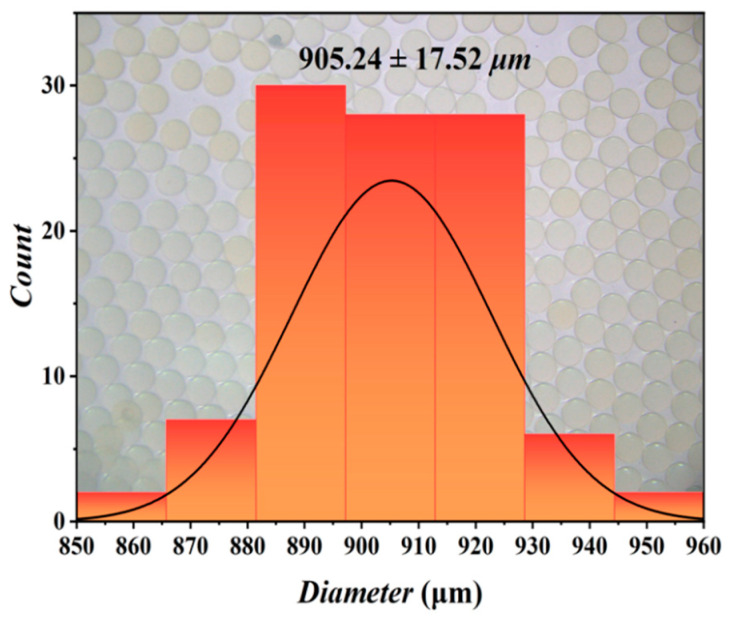
Particle size distribution of energy-containing microspheres prepared by co-flow.

**Figure 5 micromachines-16-00416-f005:**
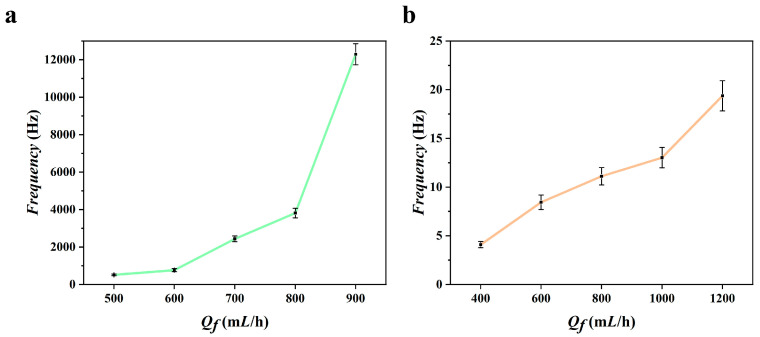
Frequency of flow-focusing and co-flow microsphere preparation. (**a**) Flow-focusing technique. (**b**) co-flow technique.

**Figure 6 micromachines-16-00416-f006:**
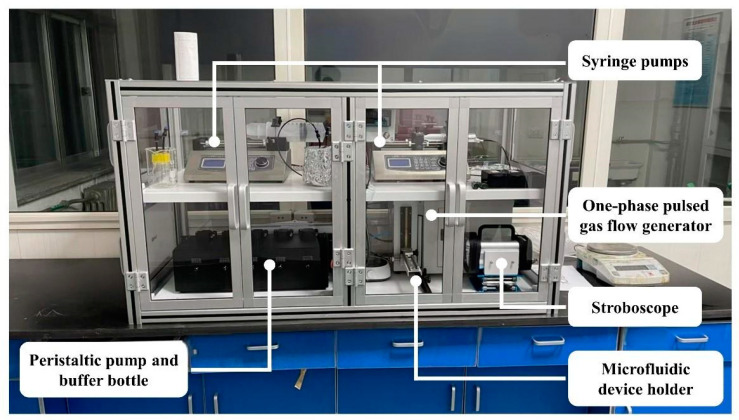
Energy-containing microdroplet preparation system.

**Table 1 micromachines-16-00416-t001:** Technologies and corresponding material systems.

Technology	Inner Phase	Driving Phase	Receiving Fluid
Flow-focusing	Photosensitive resin	PVA solution	PVA solution
Co-flow	Photosensitive resin	PVA solution	PVA solution
Pulse pneumatic printing	Sodium alginate	Pulse gas	Calcium chloride solution

**Table 2 micromachines-16-00416-t002:** Evaluation of granulation by technology.

Technology	Particle Size Distribution (μm)	Frequency	Consistency	Yield
Flow-focusing	110–305	Extremely fast	Low	Highly efficient
Co-flow	270–905	Fast	High	Inefficient
Pulse pneumatic printing	860–2550	Low-frequency Controllable	High	Efficient

## Data Availability

The original contributions presented in the study are included in the article/Appendix A, further inquiries can be directed to the corresponding author.

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
