# Peer review of "High-Quality Preparation of Energy-Containing Microspheres with Cross-Scale Particle Size"

_micromachines, 2025, doi:10.3390/mi16040416_

Round 1

Reviewer 1 Report

Comments and Suggestions for Authors

This manuscript presents a microfluidic granulation system for producing energy-containing microspheres by integrating flow-focusing, co-flow, and pulsed pneumatic printing techniques. The authors address a key challenge in the field—the limited range of particle size control that has previously restricted the broader application of microfluidic synthesis for energy-containing materials. The study systematically compares the three techniques, providing a strong experimental foundation. However, several issues need to be addressed before the manuscript can be considered for publication.

  1. The explanation of how the microfluidic system functions and how the three techniques are integrated should be clearer.
  2. The relationship between key parameters (flow rates, viscosity, surface tension) and microsphere formation needs further theoretical background and discussion.
  3. The manuscript does not address the fabrication process of the microfluidic device, which is crucial for reproducibility.
  4. Droplet and microsphere size are highly dependent on channel geometry and other design parameters, yet the paper does not provide meaningful insights into this aspect.
  5. A more explicit comparison with similar studies in terms of performance, size control, and efficiency would strengthen the impact of the findings.
  6. The stability of the microspheres under different storage conditions remains unexplored and should be discussed.
Comments on the Quality of English Language
  1. Minor grammatical inconsistencies are present throughout the text. A thorough proofreading would improve clarity and readability.

Author Response

Comments 1: The explanation of how the microfluidic system functions and how the three techniques are integrated should be clearer.

Response 1:Thank you for pointing this out. We agree with this comment. Our point-by-point replies are provided as follows:

  The flow-focusing device primarily consists of an inner stainless steel needle and a focusing chamber, as illustrated in Fig. S1(a). The needle is inserted into the chamber, with its tip aligned precisely with the focusing orifice at the chamber’s base. A higher-velocity focusing fluid, introduced from the periphery, exerts radial compressive forces on the inner-phase fluid, shaping it into a slender conical jet within the focusing zone. Under the

combined action of shear forces and surface tension, the inner conical jet undergoes hydrodynamic instability downstream, leading to its breakup into uniform droplets. These droplets are subsequently transported through a receiving conduit, where UV-induced photopolymerization solidifies them into monodisperse microspheres.

Some statements have been added in the revised Supporting Information as follows:

” The flow-focusing device primarily consists of an inner stainless steel needle and a focusing chamber, as illustrated in Fig. S1(b). The flow-focusing microchannel employs a 24G/25G inner needle paired with a focusing chamber featuring a focusing orifice of 500μm/300μm in diameter. “

The coaxial flow device is primarily composed of an inner stainless steel needle and a coaxial focusing chamber, as depicted in Fig. S1(c). The high concentricity between the needle and the chamber ensures that the focusing fluid exerts stable shear forces on the inner-phase fluid, enabling uniform droplet generation. At the outlet of the receiving channel, the droplets undergo UV-induced photopolymerization, ultimately forming monodisperse microspheres.

Some statements have been added in the revised Supporting Information as follows:

“A 20G or 22G inner needle is integrated with a focusing chamber channel of 1700 μm in diameter.”

The pneumatic printing device is primarily composed of a stainless steel needle, a continuous liquid supply unit, and a pulsed gas supply module, as illustrated in Fig. 3.1c. The inner capillary is supplied with a continuous fluid flow by a precision syringe pump, while the outer capillary delivers pulsed airflow via the controlled opening and closing of a solenoid valve. Under the shear forces induced by the pulsed airflow, droplets are generated at the capillary tip. The solenoid valve enables precise adjustment of the pulsed airflow frequency and pulse width, ensuring consistent droplet formation. Subsequent chemical crosslinking solidification transforms the droplets into monodisperse microspheres.

Some statements have been added in the revised Supporting Information as follows:

“The pneumatic printing device is primarily composed of a stainless steel needle, The microchannel utilizes a 20G/22G inner needle and a focusing chamber channel with a diameter of 1900 μm/1710 μm.”

To address the issue of how to integrate these three technologies, we have revised and refined the first paragraph of the Systematic Technology Integration Analysis on line 245 in Section 4. (Systematic Technical Integration Analysis)

“4.1. Core Nozzle Configurations

The three technologies—flow-focusing, coaxial flow-focusing, and pneumatic printing—share identical liquid supply components but differ primarily in their fluid outlet configurations. In flow-focusing, the inner needle aligns directly with a narrow focusing orifice in the focusing chamber. In contrast, coaxial flow-focusing and pneumatic printing employ concentric alignment of the inner needle with a wider focusing chamber outlet.

4.2. Focusing Phase Mechanisms

The focusing phase can be categorized as either a homogeneous liquid phase (liquid-driven) or a heterogeneous gas phase (gas-driven). Liquid-driven systems rely on stable liquid-liquid interfacial shear forces, which are further divided into flow-focusing and 

coaxial flow-focusing modes. Pneumatic printing, however, utilizes gas-driven shear, where gas-liquid interfacial interactions exhibit lower stability and weaker, more fluctuating shear forces compared to liquid-liquid systems. Notably, liquid-driven systems may introduce trace contamination due to material interactions between phases, whereas gas-driven systems minimize contamination by avoiding direct contact between the inner fluid and reactive media.

4.3. Droplet Formation Mechanisms

All three technologies require external focusing forces to fragment the inner fluid into droplets:

Flow-focusing: Droplets form via hydrodynamic instability of a slender jet in the focusing zone.

Coaxial flow-focusing: Droplets are generated through steady shear-driven fragmentation at the inner needle outlet.

Pneumatic printing: Droplets form at the capillary tip via gas-driven shear and gravitational forces.

4.4. Comparative Performance

The differences in nozzle geometry, focusing phase, and fragmentation dynamics lead to distinct droplet characteristics under equivalent flow parameters:

Flow-focusing systems: Higher interfacial shear forces (due to velocity gradients between phases) result in higher droplet breakup frequencies and smaller sizes, albeit with reduced monodispersity. This technology is ideal for high-throughput production of submicron droplets.

Coaxial flow-focusing systems: Lower velocity gradients yield smaller interfacial shear forces, producing larger droplets with superior monodispersity, broader parameter tolerance, and isotropic properties. This method is optimal for applications requiring uniform droplet morphology.

Pneumatic printing systems: Gas-liquid interfacial instability and weaker shear forces, combined with gravitational effects, generate larger droplets with minimal contamination. This approach is suited for niche applications demanding large droplets or ultrapure formulations.”

Comments 2: The relationship between key parameters (flow rates, viscosity, surface tension) and microsphere formation needs further theoretical background and discussion.

Response 2: Thank you for pointing this out. We agree with this comment. We have carefully considered this issue and provide the following explanation:

The formation of monodisperse microspheres is influenced by multiple factors, requiring careful optimization of material properties and flow parameters to ensure stable generation1,2. Flow rate directly governs shear forces and the droplet formation process: higher focusing flow rates typically result in smaller microspheres due to increased shear forces accelerating droplet breakup, whereas lower focusing flow rates yield larger microspheres. Similarly, higher dispersed-phase flow rates lead to larger microspheres by increasing fluid volume at the nozzle outlet, while lower dispersed-phase rates produce smaller droplets. Experimental results in this study corroborate these trends3.

Viscosity, which determines fluidic resistance and droplet formation dynamics, plays a critical role4. Our technique accommodates fluids with viscosities spanning 1–5000 cP, demonstrating broad compatibility. Surface tension further impacts droplet stability and morphology5. The focusing phase, containing 2 wt% polyvinyl alcohol (PVA), reduces surface tension to promote spherical droplet formation, minimize coalescence, and enhance microsphere uniformity.

These three factors—flow rate, viscosity, and surface tension—are intricately interconnected. Flow rate and viscosity jointly dictate shear forces: higher shear forces accelerate droplet breakup, leading to smaller microspheres. Surface tension governs capillary forces, with stronger capillary forces stabilizing spherical droplet shapes. Additionally, the balance between flow rate and viscosity determines flow stability, where steady flow conditions are essential for achieving uniform microspheres.

We sincerely appreciate the valuable suggestions regarding the relationship between material properties and droplet generation. However, the primary objective of this study is to explore the optimization and integration of microfluidic technology for specific applications, rather than conducting an in-depth investigation into the interplay between material characteristics and droplet formation. Consequently, our discussion in this section remains concise. Future studies could delve deeper into this direction to further elucidate the underlying mechanisms.

Comments 3: The manuscript does not address the fabrication process of the microfluidic device, which is crucial for reproducibility.

Response 3: Thank you for pointing this out. We agree with this comment. We have carefully considered this issue and provide the following explanation:

The microfluidic devices used in this study were fabricated using well-established Computer Numerical Control (CNC) machining protocols, which are widely adopted in microfluidics research.

Some statements have been added in the revised Supporting Information as follows:

1. Mold Design: CAD models of the microchannel networks (e.g., focusing orifices, inner-phase and focusing-phase inlets) were generated based on hydrodynamic simulations and extensive practical experience.

2.Material Selection: Components were machined from aerospace-grade aluminum, which offers high precision, durability, and compatibility with microfluidic applications.

3.Device Assembly: The individual components were assembled with high precision to ensure seamless integration and reliable operation.”

Comments 4: Droplet and microsphere size are highly dependent on channel geometry and other design parameters, yet the paper does not provide meaningful insights into this aspect.

Response 4: Thank you for pointing this out. We agree with this comment. We have carefully considered this issue and provide the following explanation:

  1. Microchannel Design for Flow-Focusing Technology

The flow-focusing microchannel employs a 24G/25G inner needle paired with a focusing chamber featuring a focusing orifice of 500 μm/300 μm in diameter. The high concentricity between the inner needle and the focusing orifice amplifies the shear forces exerted by the focusing fluid, enabling the inner-phase fluid to form a jet downstream. This jet undergoes hydrodynamic instability and breaks up into monodisperse droplets with reduced dimensions.

  1. Microchannel Design for Coaxial Flow-Focusing Technology

In the coaxial flow-focusing system, a 20G or 22G inner needle is integrated with a focusing chamber channel of 1700 μm in diameter. The high concentricity between the coaxial needles ensures stable shear forces from the focusing fluid, facilitating uniform droplet generation through controlled fragmentation of the inner-phase fluid at the needle outlet.

  1. Microchannel Design for Pneumatic Printing Technology

The pneumatic printing microchannel utilizes a 20G/22G inner needle and a focusing chamber channel with a diameter of 1900 μm/1710 μm. This configuration allows the inner-phase fluid to form uniform, larger-sized droplets under gas-driven shear forces, leveraging pulsed airflow modulation for precise control over droplet formation dynamics.

Comments 5: A more explicit comparison with similar studies in terms of performance, size control, and efficiency would strengthen the impact of the findings.

Response 5: Thank you for pointing this out. We agree with this comment. We have sought out relevant studies to compare and highlight the advantages of our technology.

Table R1 Comparison of Different Microsphere Preparation Techniques

Manufacturing Technology

Main Materials

Droplet Uniformity

Average Particle Size

Yield

Reproducibility

Reference

Interfacial Polymerization

Melamine

Poor

3.8μm

High

Poor

6

Spray Drying

Chitosan

Fair

10.27 ± 1.05 μm

High

Good

7

Microfluidic Chip

1 wt% Sodium Alginate

Good

100.7 μm

Low

Poor

8

Inkjet Printing

0.5 wt% Sodium Alginate

Fair

19.5 μm

Low

Good

9

Superior Droplet Uniformity and Size Control:

Coaxial Flow-Focusing: With a 1700 μm channel and 20G/22G needles, this technology generates droplets in the 270–905 μm range (CV=1.9%), surpassing spray-drying methods (Reference 7: CV ~10%) in both size tunability and reproducibility.

Enhanced Efficiency and Scalability:

Throughput: Flow-focusing and Pneumatic Printing and coaxial systems achieve a high yield (5–100mL/h) while maintaining precision, outperforming traditional microfluidic chips (Reference 8: low yield) and spray drying (Reference 7: high yield but poor uniformity).

Multifunctional Integration:

Unlike single-mode systems (e.g. Reference 6,9), our platform integrates UV photopolymerization, chemical crosslinking, and coaxial encapsulation, enabling tailored microspheres (core-shell, Janus) with diverse compositions and functionalities.

Comments 6: The stability of the microspheres under different storage conditions remains unexplored and should be discussed.

Response 6: We sincerely appreciate the reviewer’s valuable suggestion regarding the stability of microspheres under different storage conditions.We fully acknowledge that stability analysis is a critical aspect for evaluating the practical applicability of functional microspheres, especially in energy-related formulations. However, the primary objective of this study is to establish a versatile microfluidic platform for simulating the preparation of energetic microspheres with controllable sizes and morphologies, focusing on technology optimization (e.g., nozzle design, shear force modulation) and process reproducibility.

The stability of microspheres inherently depends on the specific material properties (e.g., chemical composition, crosslinking density) rather than the fabrication technology itself. In this work, we intentionally selected model materials (e.g., sodium alginate, PVA) to validate the feasibility of our three technologies in generating monodisperse droplets, as actual energetic materials may introduce safety risks and experimental complexity at this exploratory stage. Consequently, stability experiments under varying storage conditions (e.g., temperature, humidity) would not directly reflect the capabilities of our proposed technologies but instead pertain to material-specific characterization, which falls outside the current scope.

We agree with the reviewer that future studies should rigorously investigate the stability of energy-loaded microspheres (e.g., containing nitrocellulose or RDX) tailored for real-world applications. Such work will leverage the foundational parameters (e.g., droplet size, shell thickness) established in this study to design microspheres with enhanced stability. This follow-up research is already planned as part of our ongoing collaboration with industrial partners.

Thank you again for this insightful comment, which will guide our future efforts in bridging technological innovation with practical implementation.

4. Response to Comments on the Quality of English Language

Point 1: Minor grammatical inconsistencies are present throughout the text. A thorough proofreading would improve clarity and readability.

Response 1:  We sincerely appreciate the reviewer’s meticulous attention to detail and valuable feedback regarding the minor grammatical inconsistencies in the manuscript. We acknowledge that precise language is essential for conveying scientific rigor and ensuring readability. The manuscript has been thoroughly reviewed and revised to eliminate inconsistencies in tense, article usage, and sentence structure.

Original: “LTD”
Revised (Line 8) : “Ltd.”

Original: “Hefei Create Micro New Material Technology Co., Ltd.; programmable pulse gas printer, Hefei Create Micro New Material Technology Co., Ltd.;”
Revised (Line 80): “Hefei Zhongke Create-Micro Technology Co., Ltd.; programmable pulse gas printer, Hefei Zhongke Create-Micro Technology Co., Ltd.; “

Original: “energy-laden containing materials..”
Revised (Line 65): “energy-containing materials.”

5. Additional clarifications

[Here, mention any other clarifications you would like to provide to the journal editor/reviewer.]

Reference:

(1) Zhang, L.; Li, W.; Wei, L.; Zhao, Y.; Qiu, Y.; Liu, H.; Huang, C.; Huang, J. Optimizing the Production of Hydrogel Microspheres Using Microfluidic Chips: The Influence of Surface Treatment on Droplet Formation Mechanism. Langmuir 2023, 39 (39), 13932-13945.

(2) Luo, G.; Du, L.; Wang, Y.; Lu, Y.; Xu, J. Controllable preparation of particles with microfluidics. Particuology 2011, 9 (6), 545-558.

(3) Schneider, T.; Chapman, G. H.; Häfeli, U. O. Effects of chemical and physical parameters in the generation of microspheres by hydrodynamic flow focusing. Colloids and Surfaces B: Biointerfaces 2011, 87 (2), 361-368.

(4) Davarcı, F.; Turan, D.; Ozcelik, B.; Poncelet, D. The influence of solution viscosities and surface tension on calcium-alginate microbead formation using dripping technique. Food Hydrocolloids 2017, 62, 119-127.

(5) Ashikhmin, A.; Piskunov, M.; Kochkin, D.; Ronshin, F.; Chen, L. Droplet Microfluidic Method for Estimating the Dynamic Interfacial Tension of Ion-Crosslinked Sodium Alginate Microspheres. Langmuir 2024, 40 (30), 15906-15917.

(6) Zhang, Z.; Zhang, Z.; Chang, T.; Wang, J.; Wang, X.; Zhou, G. Phase change material microcapsules with melamine resin shell via cellulose nanocrystal stabilized Pickering emulsion in-situ polymerization. Chemical Engineering Journal 2022, 428, 131164.

(7) Zhang, Z. l.; Li, L. j.; Sun, D.; Wang, M.; Shi, J. r.; Yang, D.; Wang, L. h.; Zou, S. c. Preparation and properties of chitosan-based microspheres by spray drying. Food Science & Nutrition 2020, 8 (4), 1933-1941.

(8) Zhang, P.; Su, D.; Shen, X.; Xie, P.; Jiang, Z. Preparation and evaluation of microcapsules of sodium alginate based on microfluidic technology. Food Hydrocolloids 2024, 154, 110113.

(9) Chung, J. H. Y.; Naficy, S.; Wallace, G. G.; Naficy, S.; O'Leary, S. Inkjet-Printed Alginate Microspheres as Additional Drug Carriers for Injectable Hydrogels. Advances in Polymer Technology 2016, 35 (4), 439-446.

Reviewer 2 Report

Comments and Suggestions for Authors

The preparation of microspheres, especially the large - range regulation of uniformity and size, holds significant research value. The research in this paper has yielded some interesting results, but some aspects are still not clear enough:

1. There is a lack of detailed schematic and physical diagrams of the experimental setup and the structure of the microfluidic chip.
2. The rationale for the selected liquid flow rates in Figures 2 and 3 needs to be explained.
3. The formula in line 215 is neither explained in terms of its source nor numbered.
4. The results in Figure 5 do not indicate the number of repetitions, nor are error bars provided.

Author Response

Comments 1: There is a lack of detailed schematic and physical diagrams of the experimental setup and the structure of the microfluidic chip.

Response 1: Thank you for pointing this out. We agree with this comment. We have carefully considered this issue and supplemented the revised Supporting Information with detailed schematics and physical diagrams of the experimental setup and microfluidic chip structure (Figs. S1(a-c)), which will help readers better understand our experimental design.

Comments 2: The rationale for the selected liquid flow rates in Figures 2 and 3 needs to be explained.

Response 2: Thank you for pointing this out. We agree with this comment. We have carefully considered this issue and provide the following explanation:

  1. Flow-Focusing Technology (Figs. 2a, 3a)

Flow Rate Range:

  • Inner Liquid (Dispersed Phase): Fixed at 30 mL/h (light-sensitive resin suspension).
  • Focusing Liquid (Continuous Phase): Varied from 500 mL/h to 900 mL/h (2 wt% PVA solution).

Rationale:

Hydrodynamic Instability and Shear Forces:

The flow rate ratio (Qi/Qf ​) governs the interfacial shear forces that dominate droplet breakup. Higher driving flow rates (Qf​) amplify shear forces, accelerating jet instability and reducing droplet size. This aligns with the Weber number (We) theory, where We=  (with ρ: density, v: velocity, l: characteristic length, σ: surface tension). Increased Qf​ elevates v, thereby increasing We and promoting finer droplet fragmentation.

Empirical Optimization:

Preliminary experiments identified 500–900 mL/h as the critical range for stable cone-jet formation. Below 500 mL/h, insufficient shear forces caused irregular dripping; above 900 mL/h, turbulence disrupted monodispersity.

Microsphere diameter decreased from 315 μm to 110 μm as Qf​ increased, demonstrating precise size control via shear modulation (Fig. 3a).

  1. Coaxial Flow-Focusing Technology (Figs. 2b, 3b)

Flow Rate Range:

  • Inner Liquid: Fixed at 5 mL/h (light-sensitive resin suspension).
  • Focusing Liquid: Varied from 300 mL/h to 1400 mL/h (2 wt% PVA solution).

Rationale:

Shear Force Balance:

Unlike flow-focusing, coaxial systems lack a focusing orifice1, relying solely on shear forces at the coaxial interface. The selected Qf​ range (300–1400 mL/h) ensured laminar flow conditions (Re<2000) to avoid turbulence while maintaining sufficient shear for droplet breakup.

Velocity Gradient Optimization:

Lower Qf (300 mL/h) minimized velocity gradients, yielding larger droplets (905 μm). Higher Qf​ (1400 mL/h) increased gradients, reducing droplet size (270 μm). This behavior is consistent with the scaling law: D~α[(Qi+Qo)/Qf]1/2Dorif 2

Monodisperse droplets (CV = 1.9%) with tunable sizes (270–905 μm) were achieved, highlighting the balance between shear forces and flow stability (Fig. 3b).

  1. Pneumatic Printing Technology (Figs. 2c, 3c)

Flow Rate Range:

  • Inner Liquid: Varied from 7.5 mL/h to 100 mL/h (2 wt% sodium alginate suspension).
  • Focusing Liquid: Fixed pulse frequency (2 Hz, compressed air).

Rationale:

Gravitational and Shear Force Coupling:

In gas-driven systems, droplet size depends on the balance between inner phase flow

rate (Qi​) and gas pulse shear. Higher Qi increases fluid volume at the nozzle tip, requiring larger droplets to maintain mass continuity (Q=πd3f/6, where f: droplet frequency).

Pulse Frequency Stability:

A fixed pulse frequency (2 Hz) eliminated timing variability, isolating Qi as the sole variable. This simplified the analysis of size-flow rate dependence.

Practical Constraints:

The upper limit (100 mL/h) was determined by the maximum viscosity tolerance of the pneumatic system.

Droplet diameter increased linearly from 860 μm to 2550 μm with Qi​, validating the scalability of gas-driven systems for large droplets (Fig. 3c).

Comments 3: The formula in line 215 is neither explained in terms of its source nor numbered.

Response 3: Thank you for pointing this out. We agree with this comment. We have carefully considered this issue and provide the following explanation:

  1. Relationship Between Droplet Volume and Breakup Frequency

In flow-focusing technology, the volume of a droplet (V) is determined by the breakup frequency (f) and the flow rate of the focused phase (Qi​). The breakup frequency corresponds to the microsphere production rate. This relationship is expressed as:

(1)

  1. Droplet Volume as a Uniform Sphere

Assuming the droplet is a uniform sphere, its volume can be calculated using the geometric formula for a sphere:

(2)

where R is the droplet radius and d is the droplet diameter.

  1. Derivation of Microsphere Production Frequency

By equating Equations (1) and (2), the production frequency (f) can be derived as:

(3)

In the revised manuscript, we have numbered and derived the relevant equations at lines 187 and 214 to more clearly present their logic and calculation processes.

Comments 4: The results in Figure 5 do not indicate the number of repetitions, nor are error bars provided.

Response 4: Thank you for pointing this out. We sincerely appreciate the reviewer’s valuable comments regarding the absence of error bars and repeated measurements in Figure 5. In response, we have conducted additional replicate experiments and added error bars to Figure 5 based on the experimental data, thereby more accurately reflecting the statistical characteristics of the data.

Some statements have been added in the revised manuscript (Line 217) as follows:

“Error bars represent standard deviation from three independent experiments.”

4. Response to Comments on the Quality of English Language

Point 1:

Response 1:    (in red)

5. Additional clarifications

[Here, mention any other clarifications you would like to provide to the journal editor/reviewer.]

References:

1.A. S. Utada et al, Monodisperse Double Emulsions Generated from a Microcapillary Device.Science308,537-541(2005).

2.Zhu Z, Si T, Xu RX. Microencapsulation of indocyanine green for potential applications in image-guided drug delivery. Lab on a Chip. 2015 Feb;15(3):646-649.

Round 2

Reviewer 2 Report

Comments and Suggestions for Authors

The revised manuscript has met the requirements for publication in the journal.